Incipient speciation, high genetic diversity, and ecological divergence in the alligator bark juniper suggest complex demographic changes during the Pleistocene

Martínez de León Rodrigo 1 2
http://orcid.org/0000-0002-2000-4741 Castellanos-Morales Gabriela 3
http://orcid.org/0000-0001-7524-7639 Moreno-Letelier Alejandra 2 amletelier@ib.unam.mx
1 Posgrado en Ciencias Biológicas, Universidad Nacional Autónoma de México , Ciudad de México , Mexico
2 Jardín Botánico, Instituto de Biología, Universidad Nacional Autónoma de México , Ciudad de México , Mexico
3 Departamento de Conservación de la Biodiversidad, Colegio de la Frontera Sur , Villahermosa , México
Casazza Gabriele
Electronic publication date: 2022 Jul 26
Publication date: 2022
Volume: 10
Electronic Location ID: e13802
Received 2022 Jan 14; Accepted 2022 Jul 7
Copyright: © 2022 Martínez de León et al.
Copyright year: 2022
Copyright holder: Martínez de León et al.
License: This is an open access article distributed under the terms of the Creative Commons Attribution License, which permits unrestricted use, distribution, reproduction and adaptation in any medium and for any purpose provided that it is properly attributed. For attribution, the original author(s), title, publication source (PeerJ) and either DOI or URL of the article must be cited.
License URL: https://creativecommons.org/licenses/by/4.0/

Keywords: Population genetics, Demographic history, Phylogeography, Ecological niche modeling, Last Glacial Maximum, Juniperus

Funding: Posgrado en Ciencias Biológicas Consejo Nacional de Ciencia y Tecnología 1085698 Dirección General de Ausuntos del Personal Académico-Universidad Nacional Autónoma de México IA201616 This paper is part of RML doctoral research at the Posgrado en Ciencias Biológicas, UNAM with a grant provided by the Consejo Nacional de Ciencia y Tecnología (CONACyT; grant number: 1085698) and funded by Dirección General de Ausuntos del Personal Académico-Universidad Nacional Autónoma de México grant number: IA201616 awarded to AML. The funders had no role in study design, data collection and analysis, decision to publish, or preparation of the manuscript.

==============================
The most recent glacial cycles of the Pleistocene affected the distribution, population sizes, and levels of genetic structure of temperate-forest species in the main Mexican mountain systems. Our objective was to investigate the effects these cycles had on the genetic structure and distribution of a dominant species of the “mexical” vegetation across North and Central America. We studied the genetic diversity of Juniperus deppeana, a conifer distributed from the Southwestern United States to the highlands of Central America. We combined information of one plastid marker and two nuclear markers to infer phylogeographic structure, genetic diversity and demographic changes. We also characterized the climatic niche for each variety to infer the plausible area of suitability during past climatic conditions and to evaluate climatic niche discontinuities along with the species distribution. We found a marked phylogeographic structure separating the populations North and South of the Isthmus of Tehuantepec, with populations to the South of this barrier forming a distinct genetic cluster corresponding to Juniperus deppeana var. gamboana. We also found signals of population expansion in the Northern genetic cluster. Ecological niche modeling results confirmed climatic niche differences and discontinuities among J. deppeana varieties and heterogeneous responses to climatic oscillations. Overall, J. deppeana’s genetic diversity has been marked by distribution shifts, population growth and secondary contact the North, and in situ permanence in the South since the last interglacial to the present. High genetic variation suggests a wide and climatically diverse distribution during climatic oscillations. We detected the existence of two main genetic clusters, supporting previous proposals that Juniperus deppeana and Juniperus gamboana may be considered two separate species.

Introduction

Some of the most important mountain systems worldwide are also well-known biodiversity hotspots (Marchese, 2015; Mutke et al., 2011; Williams et al., 2011)⁠. Mountains play a critical role in the maintenance and generation of biodiversity. This is mainly due to their unstable physiographic nature, highly variable climatic conditions and recent tectonic activity which together promote evolutionary processes in populations such as genetic differentiation, local adaptation, demographic changes, and distribution shifts (Mastretta-Yanes et al., 2015; Perrigo, Hoorn & Antonelli, 2020; Rahbek et al., 2019). In Mexico, the highest mountain systems represent a biodiversity hotspot referred to as the Madrean Pine-Oak Woodlands because of its remarkably high number of Pinus and Quercus species as the prevalent vegetation type (Marchese, 2015; Sundaram et al., 2019; Valencia, 2004). Many studies have focused on the phylogeographic patterns and demographic processes concerning plant species inhabiting both the Mexican highlands as well as the deserts and xeric shrublands that surround them (e.g. Aguirre-Planter et al., 2012, 2020; Gugger & Sugita, 2010; Mastretta-Yanes et al., 2018; Moreno-Letelier, Ortíz-Medrano & Piñero, 2013; Ornelas, Ruiz-Sánchez & Sosa, 2010; Peñaloza-Ramírez et al., 2020; Sánchez-del Pino et al., 2020; Scheinvar et al., 2017). The main phylogeographical patterns that such studies have found are an expansion of temperate flora during glaciations and restriction of xeric plants in the same periods in the Chihuahuan Desert region; latitudinal patterns of genetic diversity in arid-adapted plants; the relatively stable demographic history of plants in the Trans Mexican Volcanic Belt (TMVB) and genetic divergence between populations at each side of the Isthmus of Tehuantepec (Gutiérrez-Ortega et al., 2020). The major climatic events thought to influence these phylogeographic patterns are the aridification and cooling from the Miocene-Pliocene and the more recent climatic oscillations that characterized the Pleistocene (Hewitt, 2000; Mastretta-Yanes et al., 2015)⁠.

Mexican sclerophyllous vegetation, also called “mexical” is an interesting model for the study of these phylogeographic dynamics due to its high levels of endemism and the particular combination of temperate and more xeric elements found in it. Studies have shown that this vegetation is a remnant of the Tertiary-Tethian flora, which expanded in North America during the aridification of the Miocene and experienced range shifts during the Pleistocene (Axelrod, 1975; Van Devender, 1990; Vásquez-Cruz & Sosa, 2020). The “mexical” is characterized by dominant woody elements with adaptations to water stress and poor soils, and can be found in summer and winter rain climate regimes (Valiente-Banuet et al., 1998). Regardless of covering a small proportion of land, “mexical” has high levels of diversity and endemism and a patchy distribution along the dry slopes of the main Mexican mountain systems in the transition zone between the xeric shrublands at lower elevations and oak-pine forest at >2,500 m (Axelrod, 1975; Valiente-Banuet et al., 1998). This transition location makes “mexical” a very heterogeneous vegetation type with presumably different responses to climate change among the species in it. A conspicuous element in the “mexical” are junipers, also known as “tástcates” or “sabinos” (Juniperus L.) (Valiente-Banuet et al., 1998)⁠. Mexico is considered a center of diversity with 14–20 recognized species and intraspecific taxa, and most of them are found in the “mexical” and adjacent mixed conifer forests (Adams, 2014; Farjon & Filer, 2013; Gernandt & Pérez-de la Rosa, 2014; Zanoni & Adams, 1975, 1979).

The genus is divided into three sections, Juniperus, Caryocedrus and Sabina, the latter being the most diverse (Adams, 2014)⁠. All Mexican junipers belong to section Sabina and most of them are included in a well-recognized group known as the “serrate leaf junipers of North America”, a monophyletic clade within the section that originated in North America during the Oligocene (25.15–23 Mya) and which along with other conifer genera such as Abies and Pinus has gone through increased diversification during the Miocene (23–5 Mya) (Adams, 2000, 2014; Adams & Schwarzbach, 2013; Aguirre-Planter et al., 2012; Mao et al., 2010; Uckele et al., 2021; Willyard et al., 2007)⁠.

Here we focused on the alligator bark juniper (Juniperus deppeana Steud.), a widespread serrate leaf juniper frequent in the Southwestern pinyon-juniper woodlands in the USA and the Mexican open conifer forests from 1,900 to 3,200 m (Adams, 2014; Farjon & Filer, 2013)⁠. It comprises four varieties and up to three recognized formae (Adams, 2014; Adams & Schwarzbach, 2006; Farjon, 2005)⁠, although its status as a single species has been discussed before and its general taxonomy is considered to be unstable (Adams, 1973, 2000; Adams & Schwarzbach, 2013; Eckenwalder, 2009; Farjon, 2005; Farjon & Filer, 2013; Uckele et al., 2021). Its distribution range encompasses the main Mexican mountain ranges with varieties distributed along them in a mostly allopatric manner (Fig. 1). From the Chihuahuan Desert mountains in northern Mexico and Southwestern United States (Juniperus deppeana var. deppeana Steud), across both the Sierra Madre Occidental (SMOcc) (J. deppeana var. robusta Martínez, and J. deppeana f. zacatecensis) and the Sierra Madre Oriental (SMOr) (J. deppeana var. deppeana) to the Trans-Mexican Volcanic Belt (TMVB) in central Mexico and the Chiapan-Guatemalan highlands between Mexico and Guatemala (J. deppeana var. gamboana (Martínez) R.P. Adams). There are also some varieties and formae recorded from a small number of scattered localities with not enough individuals to be considered populations (J. deppeana var. patoniana (Martinez) Zanoni, J. deppeana f. sperryi (Correll) R.P. Adams and J. deppeana f. elongata R.P. Adams).

Figure 1 Sampled localities.

Geographical distribution of collected localietes and some highlighted geographic features in the Mexican landscape.

We aimed to describe the processes that have created the genetic diversity in the varieties of J. deppeana, as well as the effects that the glacial cycles during the Pleistocene had on its distribution range and population sizes, this in the context of the climatic heterogeneity among the mountain systems that harbor them. Our hypotheses are the following: (1) glacial oscillations affected the Southern populations of J. deppeana differently than Northern populations, due to more pronounced range shifts in the latter, therefore, we expect higher genetic diversity, lower genetic structure and evidence of population growth in the Northern populations compared to the South; (2) the broad geographic range and morphological heterogeneity could indicate ecological divergence among varieties and forms, which would in turn exhibit genetic differentiation.

We tested the first hypothesis by estimating genetic diversity and structure with both cpDNA and nrDNA markers. The second hypotesis was tested by modeling climatic niche under current and past conditions (Last Glacial Maximum and Last Interglacial), and performing a background test of niche differentiation to determine if the climatic niche is significantly different tan expected by chance.

Materials and Methods

Sample collection and DNA extraction

We collected leaf tissue from four to five individuals from 24 localities across the whole distribution range of J. deppeana (Fig. 1), and a few individuals of J. flaccida and J. durangensis as outgroups (Collection permit no. SGPA/DGGFS/712/1768/16 issued by SEMARNAT). Samples were sorted by variety according to Adams (2014) (i.e. considering J. gamboana as a variety of J. deppeana) (Table 1). Nevertheless, variety deppeana has a discontinuous distribution and individuals from its northern range have been proposed to belong to a different variety (i.e. Juniperus deppeana var. pachyphlaea (Torr.) Martinez) (Farjon & Filer, 2013)⁠; therefore, we decided to separate the data into two subsets (North and South) to further investigate the division into two varieties (north and south). For phylogeographic studies, it is more informative to have a good representation of genetic variation in a large geographic area (more populations) than within populations. Therefore, as our aim is to describe the processes that have originated the genetic diversity in the varieties of J. deppeana, we favored sampling a high number of localities over the number of individuals for each locality (Aguirre-Liguori et al., 2020)⁠.

Table 1 Geographic information of each locality collected of Juniperus deppeana and the number of individuals collected per locality (N).

Locality	N	Latitude	Longitude	Variety	
Payson	5	37.3191	−111.4058	deppeana (North)	
Chiricahua	4	31.9722	−109.3477	deppeana (North)	
Mt. Lemmon	4	32.3726	−110.6930	deppeana (North)	
Los Lirios	4	25.3784	−100.6445	deppeana (North)	
Nuevo León	4	25.3784	−100.5123	deppeana (North)	
Tlalpujahua	4	19.8048	−100.1626	deppeana (South)	
Amozoc	4	19.0573	−98.0222	deppeana (South)	
Virreyes	4	19.0573	−97.6423	deppeana (South)	
Epazoyucan	4	20.0594	−989.6060	deppeana (South)	
La Palmita	4	23.9340	−104.8956	robusta	
El Salto	4	23.7781	−105.7297	robusta	
Buenos Aires	4	23.6885	−105.7297	robusta	
Lago Arareko	5	27.7110	−107.5951	robusta	
Rejogochi	3	27.4121	−107.4951	robusta	
Basaseachi	4	28.2210	−108.1822	robusta	
La Junta	4	28.4097	−107.4060	robusta	
Madera	4	29.1408	−107.9435	robusta	
Sombrerete	4	23.6685	−103.6794	zacatecensis	
Cargadero	4	22.7551	−103.0980	zacatecensis	
Sierra Fría	3	22.1599	−102.5481	zacatecensis	
La Congoja	2	22.1644	−102.5654	zacatecensis	
San Crisóbal	5	16.6483	−92.5458	gamboana	
Teopisca	5	16.5706	−92.4997	gamboana	
Amatenango	5	16.5096	−92.3701	gamboana	

Three genetic markers were amplified and further sequenced for all individuals, including samples from J. durangensis and J. flaccida: Myb transcription factor (Myb) and the second intron of the gene LEAFY (Lfy) for nuclear representation and trnL-trnF spacer from cpDNA. These markers were selected because they have shown relatively high variability in other studied Juniperus species and due to their single-copy nature (Li et al., 2011; Moreno-Letelier, Mastretta-Yanes & Barraclough, 2014; Willson, Manos & Jacson, 2008).

Total genomic DNA was extracted from collected leaf tissue using a 2× cetyltrimethylammonium bromide (CTAB) modified protocol (Doyle & Doyle, 1987)⁠ and DNA integrity was evaluated with a 1.0% agarose electrophoresis gel. PCR amplifications were carried out using 5 µl (10–30 ng) of DNA in 12.5 µl of PCR Master mix (Promega, Madison, WI, USA), 0.5 µl of each primer and nuclease-free water was added for a total reaction volume of 25 µl.

PCR protocols for each marker were as follows: a 30 s initial period at 98 °C, 40 cycles of 10 s at 98 °C, 40 s at 55 °C, 90 s at 72 °C and a final extension of 7 min at 72 °C for Lfy (Willson, Manos & Jacson, 2008)⁠; an initial period of 4 min at 94 °C, followed by 36 cycles of 40 s at 94 °C, 45 s at 57 °C, 2 min at 72 °C and a final extension of 10 min at 72 °C for Myb (Tsumura et al., 1997)⁠; and an initial 5 min period at 94 °C, 37 cycles of 1 min at 94 °C, 40 s at 56 °C, 1 min at 72 °C and a final extension of 7 min at 72 °C for trnL-trnF spacer (Taberlet et al., 1991)⁠. Primer sequences are detailed in Supporting Information (Table S1.1). PCR products were stored at 4 °C, purified, and reverse-forward sequenced by the Laboratorio de secuenciación genómica de la biodiversidad y la salud at the Institute of Biology (UNAM, Mexico City, Mexico).

Sequences were recovered using the UGENE Sanger Sequence toolkit (Okonechnikov, Golosova & Fursov, 2012)⁠, with a reference sequence of each marker obtained from GenBank (accession numbers: KJ365158.1 for Myb, EU277714.1 for Lfy and HM024562.1 for trnL-trnF spacer). For nuclear loci, heterozygous sites were detected and coded using the IUPAC ambiguity code. We aligned the sequences together using the MUSCLE algorithm (Edgar, 2004) ⁠ implemented in UGENE v 1.31.1 with default settings and manually inspected for alignment errors (i.e. trimming sequences at the same length, detecting missing data and misplaced indels). Haploid sequence reconstruction was further carried out for nuclear markers (i.e. Myb and Lfy) using the PHASE 2.1 algorithm (Stephens, Smith & Donnelly, 2001) ⁠ in DNAsp v. 6 (Rozas et al., 2017) ⁠ resulting in a final alignment with two sequences per individual for each nuclear marker.

Genetic structure and haplotype reconstruction

We assessed genetic structure with the Bayesian approach of the software STRUCTURE (Pritchard, Stephens & Donnelly, 2000)⁠ using a matrix of coded nuclear haplotypes previously obtained with DNAsp v. 6 (Rozas et al., 2017). K values used ranged from 1 to 10, as the number of assumed taxonomic entities with enough individuals to be considered populations, and 10 runs were performed for each K value with 100,000 iterations per run. We used a no admixture model and a correlated allele frequencies model. The no admixture model assumes genetic material of each individual comes from a single unknown population (Pritchard, Stephens & Donnelly, 2000) and therefore the probability of each individual belonging to every K group is given as the result (Hubisz et al., 2009). The correlated allele frequencies model is suggested for closely related individuals-when we expect to see similar allele frequencies-(Pritchard, Stephens & Donnelly, 2000). The optimal K value was approximated by the Evanno method (Evanno, Regnaut & Goudet, 2005)⁠ as implemented in the platform STRUCTURE HARVESTER v0.6.94 (Earl & VonHoldt, 2012)⁠ and a visual representation was obtained in STRUCTURE PLOT v2.0 (Ramasamy et al., 2014).

Summary genetic statistics were calculated using DNAsp for each locus at genetic cluster level. Summary genetic statistics include: number of haplotypes, haplotype diversity (Nei, 1987)⁠, number of segregating sites, nucleotide diversity (Nei, 1987)⁠, the average number of nucleotide differences (Tajima, 1983)⁠, Watterson’s estimator (Watterson, 1975)⁠ and Tajima’s D (Tajima, 1989). We used ‘adegenet’ package in R (Jombart, 2008; R Core Team, 2013)⁠ to evaluate an isolation by distance (IBD) scenario with a Mantel test. We estimated pairwise genetic differentiation (GST) between localities and a geographic distance matrix was approximated by calculating the euclidean distance between points using the ‘geodist’ R package (Padgham & Sumner, 2020). We inferred relationships between haplotypes by drawing haplotype networks for each locus with POPART v1.7 (Leigh & Bryant, 2015)⁠ including the sequences we obtained for both J. durangensis and J. flaccida as outgroups. We used the median-joining method described in Bandelt, Forster & Röhl (1999)⁠ and indels were coded as a fifth state regardless of their length.

To infer the phylogenetic relationship between all varieties and evaluate their monophyly we constructed a phylogenetic tree using the Multispecies coalescent model implemented in StarBEAST (Heled & Drummond, 2010) modality of the software BEAST 2.5 (Bouckaert et al., 2019) using both cpDNA and nrDNA markers. For this purpose, all sequences were grouped according to sample population, therefore instead of individuals each terminal corresponds to a group of every allele found for each locality. We implemented a Yule process as the multispecies coalescent prior and a piecewise linear and constant root population size model. Four independent replicates of 10 million generations (sampling every 1,000 generations) were combined, throwing aside the first 10% as our burn-in phase. We used Tracer 1.7 (Rambaut et al., 2018) (1) to verify the adequacy of the burn-in used and (2) to confirm the convergence of the MCMC in the replicates combined. The resulting tree with highest posterior probability was obtained with TreeAnnotator (Bouckaert et al., 2019) and posterior probability was annotated for branches and clades.

Demographic changes

To detect changes in effective population size, a multilocus Extended Bayesian Skyline Plot (EBSP) (Heled & Drummond, 2008)⁠ was generated in BEAST v2.5 (Bouckaert et al., 2019)⁠ for each of the resulting genetic clusters using the information of all three markers in a partitioned database. The most appropriate molecular substitution model was determined for each locus under the Bayesian information criterion (BIC) in jModelTest2 v2.1.10 (Darriba et al., 2012)⁠: HKY + G(0.94) for Lfy, JC69 + G(0.34) for Myb and JC69 for trnL-trnF. Two independent replicates were run with 20 million generations setting a 10% burn-in, and a strict clock model fixing the mutation rate of the trnL-trnF spacer set at 1.12 × 10−10 substitutions yr−1 as in Moreno-Letelier, Mastretta-Yanes & Barraclough (2014) and allowing the mutation rates of the nuclear genes to be estimated. The resulting logfiles of both replicates were joined together with LOGCOMBINER (Bouckaert et al., 2019)⁠ and effective sample size (ESS) values higher than 200 were verified with TRACER (Rambaut et al., 2018)⁠ as a sign of convergence.

Past and current distributions

To map the current potential distribution of each of J. deppeana’s varieties, as well as the extent of their suitability areas under past climatic conditions, i.e. Last Interglacial (LIG; ~130,000 yr ago) and the Last Glacial Maximum (LGM; ~20,000 yr ago), we use species distribution modeling (SDM) using the pseudo-absence algorithm Maxent 3.4.1 (Phillips et al., 2017)⁠ and projected them into the corresponding available climatic layers for each period.

Briefly, we compiled and curated a dataset of georeferenced records for J. deppeana with records from the Global Biodiversity Information Facility (GBIF.org on 23 July 2018. GBIF Occurrence Download: https://doi.org/10.15468/dl.ymwero) as well as collection records from the MEXU herbarium (IBUNAM, Mexico City, Mexico) and grouped them into subsets according to variety distribution. We used 10 climatic variables from the 19 bioclimatic layers available at WorldClim (at a 30 arc-sec resolution) (Hijmans et al., 2005) and tested different levels of model complexity with the R package WALLACE (Kass et al., 2018) to find the best settings to build the final models for each group. Final models were selected according to the corrected Akaike information criterion (AICc) and the AIC weight and final models were built using the average model of 20 cross-validation replicates with logistic output in MAXENT. We evaluated final models performance using the continuous Boyce Index, calculated in the R package ‘ecospat’ (Boyce et al., 2002; Di Cola et al., 2017) and the final SDM of each group was projected onto the corresponding bioclimatic layers for each period. A more detailed explanation of SDM construction including variable selection can be found in Supporting Information (Appendix S2.2).

We used bioclimatic layers for the LGM for two general circulation models (GCM), the Model for Interdisciplinary Research on Climate (MIROC-ESM) (Watanabe et al., 2011) and the Community Climate System Model (CCSM4) (Gent et al., 2011) retrieved from the Climatologies at high resolution for the Earth’s land surface areas (CHELSA; https://chelsa-climate.org/last-glacial-maximum-climate) database at a 2.5 min resolution. Bioclimatic layers used for the LIG (Otto-Bliesner et al., 2006)⁠ were retrieved from PaleoClim (http://www.paleoclim.org) (Brown et al., 2018) at a 2.5 min resolution. Final maps were constructed converting the SDM of each period (i.e. current conditions, LGM and LIG) into binary maps using a 10-percentile of the training records threshold, and areas with climatic stability between current conditions and the LGM were defined in terms of each group’s predicted suitability.

Climatic niche similarity assessment

We performed a background test analysis in ENMtools (Warren, Glor & Turelli, 2010)⁠ in order to test whether discontinuities over the distribution range of J. deppeana’s varieties are better explained by historical/random causes or by climatic niche differences. Background test is based on the comparison of niche overlap metrics such as I (Van der Vaart, 1998; Warren, Glor & Turelli, 2008)⁠ or Schoener’s D (Schoener, 1968)⁠ of two different Maxent-derived SDMs to a null distribution. Null distributions are produced over multiple comparisons of each SDM to a set of randomly produced SDM of a common background area to both taxa, thus telling if two different SDMs are more or less similar than what is expected by chance (Warren, Glor & Turelli, 2008)⁠.

We only tested comparisons between semi-adjacent groups (i.e. those pairs in which a continuous background area was possible to define), because background test is more informative about the climatic niche of the geographic distribution of the records used than the true ecological niche of the entities involved (e.g. species, subspecies, etc.) (Warren et al., 2014). We ran the following pairwise comparisons: deppeana (North) vs. deppeana (South), robusta vs. deppeana (North), zacatecensis vs. robusta, zacatecensis vs. deppeana (South) and deppeana (South) vs. gamboana. All tests were run symmetrically, meaning that every comparison resulted in two null distributions (Warren, Glor & Turelli, 2010)⁠. All comparisons were run with 100 pseudo-replicates with logistic output using the default complexity settings (i.e. all possible feature classes and a regularization multiplier set at 1) and using a general rectangular extent of −119°, −90° long and 13°, 34° lat. As the number of occurrences for each group varied considerably, we standardized comparisons by setting the number of random background points to equal the group with fewer records. For this analysis, we used the total joint datasets (i.e. training and test data combined) and background areas between groups were defined as the different biotic communities shared among groups in order to incorporate plausible information about dispersal ability to the test, as well as climatic and evolutionary affinities of each group (Warren, Glor & Turelli, 2010)⁠. We used the available biotic communities defined by Brown (2007) and the same set of non-correlated bioclimatic variables used in the mapping process described above.

Results

Genetic diversity and haplotype networks

All sequences generated in this study are deposited in the GenBank database (www.ncbi.nlm.nih.gov/genbank) under the popset numbers: 2208845402 (trnL-trnF), 2208845248 (Lfy) and 2208844880 (Myb). Total length of sequences were as follows: 302 pb (trnL-trnF), 775 pb (Lfy) and 607 pb (Myb). Polymorphism and sites with indels were highly variable among loci, as well as the number of parsimony-informative sites, being the nuclear marker Lfy the most variable (Table S3.1). The chloroplast sequences trnL-trnF showed the least genetic diversity between all loci while Lfy had the highest levels of polymorphism with up to 6.18% of polymorphic sites and 40 parsimony informative sites. Genetic diversity varied both between groups and among loci, especially for the number of haplotypes: 5 for trnL-trnF, 45 for Lfy and 27 for Myb (Table 2).

Table 2 Parameters of genetic diversity for Juniperus deppeana at the variety and genetic cluster level.

	N	s	π	θ(sec)	θw	k	h	Hd	Tajima’s D	
trnL-trnF	
var. deppeana (North)	24	2	0.00055	0.53558	0.002	0.1667	3	0.163	−1.1469	
var. deppeana (South)	15	2	0.00128	0.61509	0.002	0.381	3	0.352	−1.00161	
var. robusta	32	3	0.001	0.7449	0.002	0.3004	3	0.232	−1.37923	
var. gamboana	17	2	0.0022	0.59159	0.002	0.6618	3	0.588	0.2961	
f. zacatecensis	15	1	0.00044	0.30754	0.001	0.133	2	0.133	−1.15945	
Cluster 1	17	2	0.002	0.59159	0.002	0.6618	3	0.588	−1.45766	
Cluster 2	86	4	0.00077	0.79777	0.003	0.228		0.199	0.2961	
Overall	103	4	0.00092	0.76963	0.003	0.2754	5	0.249	−1.2688	
Lfy	
var. deppeana (North)	26	10	0.00358	2.88263	0.00407	2.5353	9	0.797	−0.40034	
var. deppeana (South)	28	16	0.00413	4.11157	0.0058	2.93122	14	0.918	−0.98882	
var. robusta	48	18	0.00341	4.05591	0.00572	2.41933	17	0.820	−1.27811	
var. gamboana	26	10	0.00204	2.62057	0.0037	1.44615	8	0.729	−1.46662	
f. zacatecensis	18	14	0.00495	4.36103	0.00615	3.5098	11	0.922	−0.74014	
Cluster 1	26	10	0.00204	2.62057	0.0037	1.44615	8	0.729	−1.46662	
Cluster 2	120	38	0.00404	7.64849	0.01079	2.86611	38	0.871	−1.92012*	
Overall	146	54	0.00462	10.2068	0.0144	3.2755	45	0.876	−2.09121*	
Myb	
var. deppeana (North)	38	7	0.00143	1.66604	0.003	0.8649	7	0.609	−1.3463	
var. deppeana (South)	26	9	0.00237	2.62050	0.004	1.4300	11	0.855	−1.48199	
var. robusta	56	16	0.00219	3.4831	0.006	1.324	16	0.784	−1.98499	
var. gamboana	26	5	0.00279	1.31029	0.002	1.6892	7	0.800	0.81956	
f. zacatecensis	26	15	0.00278	3.93086	0.007	1.6831	10	0.708	−1.98499*	
Cluster 1	26	5	0.00279	1.31029	0.002	1.6892	7	0.800	0.81956	
Cluster 2	150	28	0.00214	5.19293	0.009	1.2922	26	0.743	−2.17828**	
Overall	176	28	0.00235	5.04799	0.008	1.421	27	0.7800	−2.05019*	
Notes:

* p < 0.05.

** p < 0.01.

Genetic diversity is shown by locus. N, Number of sequences; s, segregating sites; π, nucleotide diversity; θ(sec), Watterson’s θ per sequence; θw, Watterson θ per site; k, mean number of nucleotide differences; h, number of haplotypes; Hd, Haplotype diversity and Tajima’s D.

Haplotype networks for each locus showed a similar pattern, with one widespread haplotype (two in the case of Lfy), and multiple low-frequency haplotypes connected by one or two mutational steps in a star-like shape (Fig. 2A). Interestingly, in the Lfy haplotype network, the two haplotypes with the widest distribution were shared among almost all varieties, except for most of the localities from var. gamboana which were closely related to only one of the more frequent haplotypes rather than two as in the other varieties. Also, var. gamboana showed the least low-frequency haplotypes in all the nuclear haplotype networks.

Figure 2 Haplotype network and distribution.

(A) Median joining haplotype networks for each locus. Circle sizes are relative to the number of sequences found for each haplotype and mutational steps are drawn as dashes. Grey and withe circles depict outgroups. (B) Genetic clustering and its geographic distribution. Results of the genetic clustering for K = 2 (lower) and for K = 3 (upper) and the geographic distribution of localities and their clustering results. Dots depicting localities are colored by variety: pink, var. deppeana (South); blue, var. deppeana (North); red, var. robusta; yellow, f. zacatecensis and green for var. gamboana. Pie graphs depict the percentage of individuals assigned to each genetic cluster. (C) Extended Bayesian Skyline Plots (EBSP) for genetic cluster 1, comprising localities from var. gamboana (D) Extended Bayesian Skyline Plots (EBSP) for genetic cluster 2 comprising localities from varieties deppeana (north and south), robusta, and f. zacatecensis. Ne represents the magnitude of changes in effective population size.

Phylogeographic structure and demographic changes

The STRUCTURE analysis showed support for two genetic clusters (K = 2, ∆K = 16.7680 & LnP(K) = −1,001.7300; Likelihood and ∆K values for all K values tested can be seen in Fig. S4.1), separated by the Isthmus of Tehuantepec, the first one comprises all of the northern populations (i.e. varieties deppeana, robusta and zacatecensis) and the second one is formed by the populations assigned as var. gamboana. The resulting plot for K = 3 (∆K = 9.837 & LnP(K) = −990.8100), is also shown, where the first genetic cluster is further divided into two genetic clusters that correspond each one to the northern and southern extremes of var. deppeana distribution range, and mixed clustering for the populations of var. robusta and var. zacatecensis, as well as a well-defined cluster for var. gamboana (Fig. 2B). The North-South admixture pattern detected by the clustering analysis was not supported by an IBD pattern as shown by the Mantel test results (r = 0.1863, p = 0.056) (Fig. S4.2).

The phylogenetic relationships among varieties were poorly supported with the exception of the clade which contained all J. deppeana var. gamboana populations (Fig. S5.1). However, the posterior probability of that clade can still be considered low (0.82), so we can infer that the markers used do not offer enough information to recover more robust relationships.

EBSP analysis showed a constant and abrupt demographic change for the first genetic cluster (deppeana, robusta and zacatecensis) that started somewhere between 0.12 and 0.14 Mya and continued up to the current time (Fig. 2C). This population change suggests a threefold increase in the effective population size (Ne) from 0.14 Mya to today. For the second cluster (i.e. var. gamboana) the results support the hypothesis of constant population size given that the confidence interval (95% CPD) includes zero changes in effective population size (Fig. 2D).

Current and past distributions

The selected variables for the SDM construction were: annual mean temperature (Bio1), mean diurnal range (Bio2), isothermality (Bio3), maximum temperature of the warmest month (Bio5), mean temperature or the driest quarter (Bio9), mean temperature of the coldest quarter (Bio11), annual precipitation (Bio12), precipitation seasonality (Bio15), precipitation of the driest quarter (Bio17) and precipitation of the coldest quarter (Bio19) which together account for >85% of the total variability of the whole unsorted data set. The number of final occurrence records considered varied between geographic groups. A total of 162 occurrence records for deppeana (North), 77 for deppeana (South), 109 for robusta, 32 for zacatecensis and 44 for gamboana and the optimal level of complexity varied between each group in terms of feature class and regularization multiplier (Table 3).

Table 3 Final model parameters and calibration metrics for each variety.

Group	Response class	rm	AUC training	AICc	w.AIC	BI	
var. deppeana (North)	LQH	1	0.9514	2689.1559	0.6570	0.845	
var. deppeana (South)	LQH	1	0.9370	1217.4137	0.2280	0.863	
var. robusta	LQ	0.5	0.8992	1519.5485	0.3009	0.845	
var. gamboana	LQHP	2	0.9760	652.4753	0.5117	0.853	
f. zacatecensis	L	1	0.8939	557.4725	0.1659	0.886	
Note:

Response classes: L (linear), Q (quadratic), H (hinge) and P (product). Regularization multiplier (rm). Average AUC of the model during model calibration (AUC training). Corrected Akaike Information Criterion (AICc). AIC weight (w.AIC) and the Boyce Index (BI).

In terms of position and extent, all models produced similar overlapping areas between LGM and current conditions despite the circulation model used, except for var. gamboana, in which a much smaller distribution during the LGM is predicted under the CCSM4 conditions (Fig. 2D). Moreover, both CCSM4 and MIROC-ESM suggest a major distribution shift for the northern portion of var. deppeana, changing its main distribution from the Chihuahuan Desert to the slopes and drylands in the southwestern USA, between the SMOcc and the Rocky Mountains (Fig. 3A).

Figure 3 Distribution models for the present and the Last Glacial Maximum for the varieties of Juniperus deppeana.

The current predicted distribution of J. deppeana’s varieties and projected distribution to LGM using two different GCM, CSSM4 (left) and MIROC-ESM (right). Overlapping areas between current and past distribution are considered areas of climatic stability and are presented in green. Distributions are shown in an elevation gray background ranging from lowest (lighter) to highest (darker). (A) var. deppeana (North), (B) var. deppeana (South), (C ) var. robusta, (D) var. gamboana, and (D) f. zacatecensis.

We found notable differences in the predicted LGM distribution of varieties deppeana (North), robusta and gamboana, and more subtle differences for the rest when comparing both GCMs used. While with the CCSM4 model the projected distribution of varieties deppeana (North) and robusta were more concentrated in the South Mexican Plateau and the East portion of today’s Chihuahuan Desert, with the MIROC-ESM model the predicted distribution of var. robusta for the LGM was wider in the northern part of the SMOcc with small patches surrounding today’s Chihuahuan Desert in northeast Mexico (Figs. 3A and 3C).

For the LIG (~140,000 yr ago) the projections were unable to map any suitable area for the presence of var. gamboana, while for varieties deppeana (South) and zacatecensis only a small portion of the area was mapped on the east part of the TMVB and at the lower part of the SMOcc for each group respectively (Fig. 4). For the Northernmost varieties (i.e. deppeana (North) and robusta) the potential distribution map during the LIG was concentrated in the southern portion of the Mexican plateau, the main SMOcc and a scattered distribution along the west portion of the TMVB.

Figure 4 Predicted distribution of J. deppeana’s varieties during the last interglacial (~130,000 yr ago).

Niche similarity

The overall overlap metrics were significant and consistent for most of the comparisons except the ones involving group gamboana, in which a conflict between metrics and a lack of significance in at least one of the symmetrical comparisons of each metric evaluated were found (Fig. 4). We found significant differences between three of the five comparisons evaluated: deppeana (North) vs deppeana (South), robusta vs deppeana (North), and zacatecensis vs deppeana (South)–meaning that observed values for overlap metrics, D and I, are lower than expected by chance. The relationship between zacatecensis and robusta SDMs was more complex, while the comparison of group robusta vs the background resulted in overlap metrics higher than expected by chance, when comparing zacatecensis against the same background area the overlapping metrics were lower than what is expected by chance.

Discussion

Genetic structure and genetic diversity

The levels of genetic diversity we found in J. deppeana are contrasting to previous reports for Mexican rare juniper J. blancoi. J. blancoi has an overlapping distribution range with J. deppeana along the TMVB and the SMOcc, therefore, we would expect that it has been exposed to the same climatic oscillations during the Pleistocene in these particular regions. While J. deppeana showed higher levels of genetic diversity than J. blancoi in nuclear markers (Hd = 0.743 for J. deppeana & Hd = 0.631 for J. blancoi in Myb; Moreno-Letelier, Mastretta-Yanes & Barraclough, 2014); this was the opposite for chloroplast markers to a dramatic extent (Hd = 0.860 for J. blancoi & Hd = 0.199 for J. deppeana; Mastretta-Yanes et al., 2012). This can be attributed to differences in the distribution between these two species. Contrary to J. blancoi, J. deppeana comprises a more extended and less scattered distribution range, which could facilitate gene flow in the paternal-inherited chloroplast markers and homogenize genetic variation as a consequence. Differences in genetic diversity across loci are common and recurrent in tree species, where it is expected to find relatively lower levels of chloroplast genetic diversity and high levels at nuclear loci given the low levels of mutation rate in chloroplast markers and the generally high effective population sizes (Petit & Hampe, 2006; Savolainen, Pyhäjärvi & Knürr, 2007), this was the case also for J. deppeana (Table 2). Moreover, differences in genetic diversity were also observable at the genetic cluster and geographic levels, while for nuclear markers both clusters showed somewhat similar genetic diversity (Table 2), for trnL-trnF spacer, cluster one had much higher genetic diversity, presumably following the same distribution-related reasons discussed above.

The two genetic clusters found in this work are constituted by localities at each side of the Isthmus of Tehuantepec, a lowland corridor between the Sierra Madre del Sur and the Chiapan-Guatemalan highlands that started its formation in the late Miocene to early Pliocene (~6 million years ago) (Barrier et al., 1998)⁠. The zone is a well-known barrier for several temperate species, mainly because it represents a low-suitability area in between two somewhat climatically similar mountain systems (Peterson, Soberón & Sánchez-Cordero, 1999; Zamudio-Beltrán et al., 2020) ⁠ whereas for temperate plant species it has been found to have a significant role in the genetic differentiation between populations at each side (e.g. Gutiérrez-Ortega et al., 2020; Ornelas, Ruiz-Sánchez & Sosa, 2010; Ortíz-Medrano, Moreno-Letelier & Piñero, 2008).

In this study, the populations at the east side of the Isthmus of Tehuantepec correspond to J. deppeana var. gamboana, which has been historically considered a different species mainly due to morphological differences (e.g. the number of seeds per cone) (Farjon, 2005; Martínez, 1963). More recently and according to Adams & Schwarzbach (2006, 2013), these entities should be considered varieties of the same species, following phylogenetic results. Our results contradict this taxonomic arrangement and rather support the historical arrangement (i.e. J. deppeana and J. gamboana as two different species). Moreover, it is worthy to note that Adams & Schwarzbach (2006, 2013) phylogenetic analysis used one or two individuals per species, therefore ignoring intraspecific variation. Our results, however, are the first population-level study supporting J. deppeana and J. gamboana as two different lineages with historically limited gene flow with the Isthmus of Tehuantepec acting as a barrier between them.

For the localities distributed West of the Isthmus of Tehuantepec in central-northern Mexico, our results showed no correspondence between genetic structure and the main mountain systems in Northern Mexico. Interestingly, our results suggest a slight genetic structure in a North-South pattern with admixed localities in between them along the SMOcc. Such structure is not supported by an IBD scenario as shown by the Mantel test results and could therefore obey climatic or historical causes (Fig. S4.2). This result contrasts with what is known for other Mexican junipers, like J. blancoi with a genetic structure that correspond with elevation differences and environmental factors (Mastretta-Yanes et al., 2012; Moreno-Letelier, Mastretta-Yanes & Barraclough, 2014)⁠ and J. monticola, with a marked genetic structure between the TMVB and the SMOr (Mastretta-Yanes et al., 2018)⁠. Again, differences in distribution could explain this contrast, whereas J. blancoi and J. monticola have more restricted habitat requirements (riparian and alpine environments respectively) J. deppeana has a wider distribution and is known to grow in a wider elevation range (1,900 to 3,200 m). Accordingly, species with a less restrictive elevation range are more likely to find wider suitability areas during glacial cycles and therefore to show more connectivity and less genetic structure, for restricted taxa, only elevational shifts are expected along with local extinction (Mastretta-Yanes et al., 2015).

Pollen, macrofossil and genetic evidence suggests the existence of pine-oak woodland corridors during glacial periods in most parts of the Mexican Plateau that facilitated gene flow in populations of temperate taxa across the main Mexican mountain systems (Bryson et al., 2011; Hewitt, 2000; Metcalfe, 2006)⁠. The projected SDMs for the LGM conditions in both GCM used show that the total area of suitability for the species was wider and more connected between varieties during glacial periods and supports this scenario, which could also explain the subtle genetic structure found in the populations west of the Isthmus of Tehuantepec.

Demography and distribution

Our results showed a significant increase in population sizes for the genetic cluster that comprises populations of J. deppeana from the west side of the Isthmus of Tehuantepec that took place between 0.12 and 0.14 million years ago and has continued until the present (Fig. 2C). This was also supported by Tajima’s D negative value, which usually is associated with a lack of equilibrium between genetic diversity and nucleotide diversity. In addition, nuclear haplotype networks show a star-like shape with one (or two) main haplotypes present in the majority of groups and several low-copy haplotypes with one mutational step. Both of these patterns can be explained by a reduced effect of genetic drift for fixing alleles, another consequence of population expansion (Avise, 2000).

The predicted distribution for all varieties to LGM and LIG suggests drastic changes to the distribution of the whole species in the last 140,000 years, especially in the northern part of its distribution. While we found differences in the projected distribution for the LGM between the different GCM used, both CCSM4 and MIROC-ESM projections for the varieties deppeana (North), robusta and zacatecensis projected to the LGM showed areas of high suitability in a large portion of the Mexican Central Plateau along with the dry slopes of the SMOcc and SMOr. According to palynological data, pinyon-juniper woodlands and open conifer forests dominated most of today’s Chihuahuan Desert during the LGM and until approximately 11,000 years ago, when its modern xeric composition took place and the conifer forests were displaced to higher elevations in its surrounding slopes (Gámez et al., 2017; Metcalfe, 2006). However, the plant composition in the southwestern USA area has a complex successional history from the LGM to today. Its modern composition is believed to be at least 11,000 years old, due to the transition from more cool and moist conditions to today’s dryer climate (Thompson & Anderson, 2000; Van Devender & Spaulding, 1979). Our results suggest that the distribution of J. deppeana was much wider in the Chihuahuan Desert area during the LGM. Accordingly, the current distribution of J. deppeana in the dry slopes of the SMOr in northeastern Mexico are the remnants of a more widespread pinyon-juniper woodland. Our results also suggest that its current distribution in the slopes of the northern SMOcc in southwestern USA is relatively recent (about 11,000 yrs ago) and possibly this was followed by a dispersal event from the Chihuahuan Desert populations. For variety deppeana (South) in the TMVB both MIROC-ESM and CCSM4 projections show a wider distribution during the LGM that extended from the TMVB to the Sierra Madre del Sur in southern Mexico. The overlapping areas (i.e. proposed glacial refugia) between the current and LGM predicted areas suggest that the current potential distribution of J. deppeana var. deppeana along the east portion of the TMVB has remained constant since at least the LGM to the present. In addition, the relatively low number of low-copy haplotypes found in the nuclear haplotype networks for this variety are consistent with constant population sizes. Other studies (both palynological and genetic) also support the idea that vegetation in the TMVB suffered elevation shifts during the glacial cycles with relatively little effect on effective population sizes, especially for medium to high elevation plants (Lozano-García, 1996; Lozano-García et al., 2013; Mastretta-Yanes et al., 2015; Peñaloza-Ramírez et al., 2020)⁠. For var. gamboana at the west side of the Isthmus of Tehuantepec, the EBSP showed no significant increase in effective population sizes, however, the results from the SDM projections for the LGM suggest distributional changes between the LGM to today. The latter implies a decrease in distribution from the LGM to today under the MIROC-ESM model and a distribution expansion from the LGM to today in the CCSM4 model.

For the LIG climatic conditions, the projections failed to find suitable conditions for var. gamboana and only a small area for f. zacatecensis and var. deppeana (South). Nonetheless, the projections for varieties deppeana (North) and robusta for this period showed a displacement to the South of the Mexican plateau and the SMOcc. Our results suggest that the southern Mexican plateau was consistently inhabited by J. deppeana in both glacial and interglacial periods and only became uninhabitable for the species after the LGM ca. 11,000 years ago. Projections also suggest a northward range expansion that took place between the LIG and LGM either from the populations on the SMOcc or the ones in the southern portion of the Chihuahuan Desert. Climate conditions during LIG were warmer and wetter than present ones (Berkelhammer, Insel & Stefanescu, 2021), which could help to explain the difference between past and present distribution of J. deppeana.

According to the results from our genetic analyses, the populations of the southwestern USA are slightly more genetically similar to populations in northeast Mexico (SMOr) than the populations in the SMOcc (Fig. 2B), suggesting the migration event did occur from the northern Chihuahuan Desert to the southwestern USA. Moreover, the population growth that we found in the EBSP analysis started between 0.10 and 0.15 million years ago, meaning that the dispersal in the LIG-LGM implied also steady population growth and not only a distribution shift at least in the northern range of J. deppeana’s distribution (i.e. varieties robusta, and north deppeana). Similar distribution shifts had been proposed as a response to climate changes associated to glacial-interglacial events for other temperate and arid taxa in surrounding regions; these distributional shifts are also accompanied by demographic changes and local extinction in some cases (Castellanos-Morales et al., 2016; Malpica & Ornelas, 2014; Scheinvar et al., 2017).

Niche similarity

Our results for the background test support a north-south climatic differentiation between the populations of var. deppeana as well as the climatic differentiation between each variety with their respective adjacent varieties (i.e. var. robusta and f. zacatecensis) (Fig. 5). This suggests that the distribution of these three groups are determined by climatic barriers rather than historical causes and that the differences in the distribution of the presence records in each group are not drawn at random (Warren, Glor & Turelli, 2010). This analysis also supported a complex pattern of niche overlap between var. robusta and zacatecensis in which the climatic niche of robusta is more similar to zacatecensis than what is expected by chance but not vice versa. One possible explanation for this result is that both groups share the same climatic preferences, however, only var. robusta has access to the areas of high suitability (Warren, Glor & Turelli, 2010). On the other hand, the results for the comparison between var. gamboana and deppeana (South) did not show significant support for either disparity or similarity in the climatic niches evaluated and there is no reason to believe that they occupy different climatic niches, at least in the geographic space (Fig. 5). Although this lack of significance may be due to the discordance in the number of occurrences for each group, we expected these niches to be more similar than what is expected by chance, given the genetic differentiation found in the populations at each side of the Isthmus of Tehuantepec (Fig. 1) and the results of similar studies in the Tehuantepec region (Gutiérrez-Ortega et al., 2020; Peterson, Soberón & Sánchez-Cordero, 1999).

Figure 5 Niche similarity results.

Observed niche overlap metrics, Schoener’s D (Schoener, 1968)⁠ and I (Van der Vaart, 1998; Warren, Glor & Turelli, 2008)⁠ compared to the null distributions of each group vs a common background area. Observed values for each comparison are represented by yellow lines. Significance was approximated by 100 pseudoreplicates. D: niches are more different than what is expected by chance; and S: niches are more similar than what is expected by chance. depN : var. deppeana (North); depS: var. deppeana (South); rob: var. robusta; zac: var. zacatecensis; gamb: var. gamboana.

Interestingly, the genetic groups recovered in the clustering analysis West to the Isthmus of Tehuantepec (Fig. 2B) do not reflect a climatic differentiation scenario, shown by the background test results; and rather suggest a North-South mix of genetic diversity involving the two portions of var. deppeana, var. robusta and zacatecensis. Other common “mexical” species with similar distributions, such as the North American pinyon pines (Pinus L. subsection Cembroides) show similar patterns of climatic differentiation in the geographic space (Ortíz-Medrano et al., 2016). Nevertheless, since the methods used (i.e. ENM based approaches) only compare groups in the geographic space and do not incorporate explicit information about climatic tolerances of each group, it is not possible to infer a process of niche divergence or tolerance evolution from these results (Broennimann et al., 2012; Brown & Carnaval, 2019; Warren et al., 2014). Our results, however, highlight the climatic discontinuities that exist in J. deppeana’s distribution range in the main mountains of Mexico (i.e. TMVB, SMOcc, and SMOr) and show the climatic heterogeneity that this species exhibits in the geographic space. Future efforts should focus on the extent to which these climatic differences can shape the distribution of genetic diversity and how well they could predict genetic structure.

Juniperus deppeana and the mexical

The changes in distribution of plant communities from the Southwestern United States and Northern Mexico during the Pleistocene have been observed in paleoecological records (Van Devender, 1990; Lozano-Garca, Ortega-Guerrero & Sosa-Nájera, 2002; Metcalfe, 2006; Holmgren, Norris & Betancourt, 2007). However, little is known about the effects these changes had on individual species. The few studies that exist on the vegetation from the “mexical” and the Chihuahuan Desert suggest Southward shifts of all populations, and generally a more extensive distribution range for species with temperate affinity (Metcalfe et al., 2002; Gugger et al., 2011; Moreno-Letelier, Ortíz-Medrano & Piñero, 2013; Scheinvar et al., 2017; Loera, Ickert-Bond & Sosa, 2017; Scheinvar et al., 2020; Vásquez-Cruz & Sosa, 2020). Another interesting pattern is that the southward range expansion of many species seems to have reached the TMVB, which explains the affinity of the vegetation of the Tehuacán-Cuicatlán valley with the Chihuahuan Desert and the mexical (Valiente-Banuet et al., 1998; Loera, Ickert-Bond & Sosa, 2017; Scheinvar et al., 2020). Our results also support this broad pattern. All northern varieties of J. deppeana (deppeana North, zacatecensis and robusta) increased their distribution range and shifted southward, reaching the TMVB; meanwhile, the more “tropical” J. gamboana showed little change. These distribution changes are also supported by the amount of genetic variation and negative values of Tajima’s D found in the varieties North of the Isthmus of Tehuantepec. It has already been reported that the communities of the Chihuahuan Desert and adjacent areas have changed a lot since de Miocene (Gámez et al., 2017; Scheinvar et al., 2020; Zavala-Hurtado & Jiménez, 2020). One of the most dramatic changes seems to be the shift from a dominant mexical vegetation (open pine-juniper woodland and chaparral) during the glacial periods to a xeric scrub during interglacials and the Holocene (Thompson & Anderson, 2000; Betancourt et al., 2001; Holmgren, Betancourt & Rylander, 2006; Albert, 2015; Lozano-García et al., 2015). These cyclical changes in distribution, together with the environmental heterogeneity of the area, are surely the cause of the high levels of diversity and endemism found among arid-adapted vegetation in Northern Mexico and Southwestern United States (Axelrod, 1975; Rzedowski, 2005; Villareal-Quintanilla et al., 2017; Zavala-Hurtado & Jiménez, 2020). Moreover, results also suggest that the environmental heterogeneity in the woody elements of mexical may be a determinant factor in its establishment, expansion/contraction and conservation.

Conclusions

Our results from genetic and ecological niche analyses suggest the existence of two different lineages (J. deppeana and J. gamboana) with a complex evolutionary history marked by distributional shifts, demographic changes and a process of genetic divergence driven mainly by the existence of a geographic barrier (Isthmus of Tehuantepec). Following these results, the general taxonomy of J. deppeana and its varieties should be revisited using a more integral view and taking into consideration intraspecific variation at genetic level. It is possible, given the climatic heterogeneity occupied by J. deppeana’s populations, that other causes beyond the historical ones could explain the genetic structure found in populations in Mexico like contemporary gene flow or human influences, but these hypotheses still have to be tested. It is also worthy to note that while the genetic data showed an important demographic expansion, the projected distributions for LGM and LIG showed that this expansion possibly was not homogeneous across the species range. Therefore future studies should explore and test more complex demographic and migration scenarios.

Supplemental Information

Supplemental Information 1 Supplementary information: data, methods and results.

Click here for additional data file.

Supplemental Information 2 Sequence alignment of lyf gene.

Click here for additional data file.

Supplemental Information 3 Sequence alignment of myb gene.

Click here for additional data file.

Supplemental Information 4 Sequence alignment for the trnL-trnF spacer.

Click here for additional data file.

We thank PhD Lidia I. Cabrera, MSc Nelly M. López and MSc Laura Márquez for their valuable technical support in the DNA extraction and molecular markers’ amplification at the Laboratorio de secuenciación genómica de la biodiversidad y la salud and the Laboratorio de sistemática molecular (botánica) at the Instituto de Biología, UNAM. We also thank David Gernandt, Lev O. Jardón, Alicia Mastretta-Yanes, and Ivalú Cacho for their comments and suggestions in the development of this research. We thank Rodolfo Salas-Lizana, Verónica Moreno and Enrique Scheinvar for their aid in field work.

Additional Information and Declarations

Competing Interests

Author Contributions

Field Study Permissions

Data Availability

The authors declare that they have no competing interests.

Rodrigo Martínez de León conceived and designed the experiments, performed the experiments, analyzed the data, prepared figures and/or tables, authored or reviewed drafts of the article, and approved the final draft.

Gabriela Castellanos-Morales analyzed the data, authored or reviewed drafts of the article, and approved the final draft.

Alejandra Moreno-Letelier conceived and designed the experiments, analyzed the data, authored or reviewed drafts of the article, and approved the final draft.

The following information was supplied relating to field study approvals (i.e., approving body and any reference numbers):

Plant samples were collected with the permission of Secretaría de Medio Ambiente y Recursos Naturales of the Mexican government (No. SGPA/DGGFS/712/1768/16).

The following information was supplied regarding data availability:

The sequences generated in this study are available at GenBank: HM024562.1 (trnL-trnF), EU277714.1 (Lfy), and KJ365158.1 (Myb).

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
