# Peer review of "Incipient speciation, high genetic diversity, and ecological divergence in the alligator bark juniper suggest complex demographic changes during the Pleistocene"

_PeerJ, doi:10.7717/peerj.13802_

## Round 0.1 · original submission · Major Revisions

Dear Dr. Moreno-Letelier,

The reviewers find your work sound, well written, and worthy of publication. They, however, suggest some improvements to make the manuscript clearer and more interesting for a wide audience. So, I encourage you to improve the manuscript according to the tips of reviewers. Please, respond point-to-point to the comments of reviewers to speed up the process of revision.

Once again, thank you for submitting your manuscript to PeerJ and we look forward to receiving your revision.

Sincerely,
Gabriele Casazza

Reviewer 1 ·

Basic reporting

No comment

Experimental design

No comment

Validity of the findings

No comment

Additional comments

This study combines phylogeography using 3 sequence data sets and species distribution modelling to investigate the genetic structure and diversity and niche differentiation of the Junipers deppeana complex. The analyses are thorough and mostly well presented. However, some work needs to be done to ensure that the main conclusions are fully supported by the results. In addition, the modelling and niche differentiation results could be simplified to ease understanding. Please see my other comments and suggestions below:
Line 32-33 I am not convinced that a north-south clinal structure has been detected. The mantel test (appendix 4) figure in the supp materials includes all populations and is still not significant (Line 282). What other evidence is there for a clinal structure?
Line 88 I suggest replacing ‘taxonomy has been unstable’ with ‘taxonomy is considered unstable’ or ‘taxonomy is unresolved with wide morphological variation’.
Line 113 Why were a few individuals J. flaccida and J. durangensis included? Are these phylogenetically related?
Line 126-127 Why were these fragments chosen for sequencing? For example, have they proved to be variable in other Junipers species?
Line 156-157 Structure is not designed for linked SNPs like sequence data. In the ‘Documentation for structure software: Version 2.3’ it says that:
‘The structure model assumes that loci are independent within populations (i.e., not in LD within populations). This assumption is likely to be violated for sequence data, or data from nonrecombining regions such as Y chromosome or mtDNA.’
And
‘However, if the data are dominated by one or a few non- or low-recombining regions, then structure could be seriously misled.’
The manual suggest that for linked sequence data ‘a safe and valid solution is to recode the haplotypes from each linked region so that haplotypes are represented as a single locus with n alleles. If there are very many haplotypes, one could group related haplotypes together.’
Can the authors please confirm if there coding method has followed this advice? I also suggest using BAPS that may improve results as it specifically implements a model for clustering DNA sequence data (see Cheng, Lu, et al. "Hierarchical and spatially explicit clustering of DNA sequences with BAPS software." Molecular biology and evolution 30.5 (2013): 1224-1228.)
Line 342 Why is the genetic diversity of J. blancoi comparable to J. deppeana? It is not obvious. Looking at the Moreno-Letelier et al., 2014 it is clear that the two species have overlapping ranges and range-wide sampling was used to estimate diversity in the Moreno-Letelier et al., 2014 paper. This may seem trivial but it is crucial for establishing one of the main points in the abstract (lines 37-38) that J. deppeana has maintained high diversity.
Figure 1 It would be best to show the distribution of nuclear and chloroplast haplotypes on a map. Using only networks it is hard to get a sense of the differentiation between var. gamboana and the other varieties and whether a cline exists in the northern part of the range.
Also, what are the white and grey circles in the haplotype networks? Are they outgroups?
Figure 2 The distribution data used for modelling should be added to the SDM maps. For var. gamboana please check if the modelled area includes its current range. Unless I am mistaken it seems to be too far to the north outside the range of var. gamboana?
Figure S4.4. What are the numbers on the x-axis? They are too large for km.
I could not find a data deposition statement.

Reviewer 2 ·

Basic reporting

The authors investigated genetic structure of Juniperus deppeana varieties (and one forma) using one cpDNA and two nDNA sequences, and conducted species distribution modeling of it. They found that one of the variety (var. gamboana) showed distinct genetic structure in nDNA and some of the varieties showed significant differences in their ecological niche. Study topic is interesting. Sampling looks enough. Although in usual only two nDNA sequences may be too small to discuss, as STRUCTURE showed clear separation between lineages, it may be enough to draw conclusion. So I think that this ms can be acceptable in JeerJ. However, I cannot agree one of the conclusion of this ms as shown in the followings.

### Major comments
I cannot agree that var. gamboana is a separate species, because there are no significant differentiation in cpDNA and ecological niche. I think that variety should be appropriate.

### Minor comments
1) L 257, Although the length of the sequences are shown in Table S3, it is better to show it also in the main text.

2) L289–290, It should be not CPD but HPD (highest posterior density).

3) L366–368, Please see my major comment. I think that distinct two varieties are appropriate. L492 is also the same.

4) Figure legend of Fig. 1, there are two "c"s exist. Moreorver, panels c and d should be invert.

5) Figure legend of Fig. 2, "d" and "e" should be invert. I think that it may be better to make the scales of each panels all the same.

Experimental design

See above.

Validity of the findings

See above.

Reviewer 3 ·

Basic reporting

The manuscript by Martinez de Léon and colleagues presents a clear study of the phylogeography Juniperus deppeana. It follows the standards of peer-reviewed articles (I did not check PeerJ’s specific ones though) and figures are OK (see detailed comments below).
Although the English used is generally good, the manuscript would benefit from a round a reading by a fluent English speaker I think, as I noticed some mistakes (but I am not native speaker myself and I might have missed other ones).
I did not find info on if or where DNA data was made available for the general public, but the DNA alignments were made available for reviewers and their quality is very good.

Experimental design

The work presented here falls within the scope of PeerJ, it is well defined and is both executed and presented rigorously.

Validity of the findings

I trust all analyses and conclusions presented in this work, and rather regret that the authors did not engage into more interpretation: they stayed very safe.

Additional comments

The manuscript by Martinez de Léon and colleagues presents a clear study of the phylogeography Juniperus deppeana. The work presented here is sound and surely worthy of publication. However, I think that some improvements could be made to make the manuscript less descriptive and help it reach a wider audience: for the moment it is centered on inferring the history of one species but I think that this kind of case study always carries important information for understanding the bigger picture, that is the history of ecosystems on Earth. Unfortunately, such a connection is not explicitly done here.

Major comments

Introduction
While the introduction is well written and easy to read, I found the presentation of « mexical » and the justification for its phylogeographic study too short. I think the authors should take the time to develop a bit on the history of this vegetation, its general ecology and its importance as an ecosystem. How is it different from other vegetation types in Mexico? What will this phylogeographic study on one juniper tell us about the history of this vegetation type?
Also, at the end of the introduction specific hypotheses about the phylogeography of Juniperus deppeana are not formulated, the only sentence is the following: ‘We hypothesized that due to the climatic differences between the distributions of its varieties, processes such as glacial oscillations in the last 2.5 My have affected each variety differently.’ This is very vague and unspecific, it could have been written for any organism in any place on Earth. The authors should explain what impacts glacial cycles might have had on the species, and how this would have differed between regions/subspecies. Without this, the manuscript is purely descriptive and only relevant to J. deppeana but we do not understand what general knowledge this study brings.

Material and methods
In the methods a lot of analyses are presented one after the other, but we are lacking an overview of why all of these methods are used. I know this is rather standard in the population genetics/phyloeography literature, but I don’t think it is a good practice. Why for example do the authors measure IBD, how is this relevant to Pleistocene glacial cycles in a complex topographic landscape? This list of analyses rather reads like the authors have tried to tick all the ‘classics’, but they might not all be necessary. I must note, however, that all of these measures are discussed, which is a very good point.

In addition to the haplotype networks presented in Fig. 1A, I am wondering why the authors did not estimate a multilocus phylogeny of the different individuals or populations? Such a phylogeny, especially if dated (which could be done exactly as for the skyline plots), would help a lot understanding the history of the species and how and when the different populations/lineages diverged from each other.

When building the SDMs with Maxent, using ‘10 non-correlated’ BIOCLIM variables (as stated in L205) seems like a dubious choice for two reasons: I doubt that there are actually 10 non-correlated variables (at least I would like it to be shown) and a more sensible choice of explanatory variables could perhaps be done based on the ecology of J. deppeana, which is not discussed. For example, is this species affected by precipitation of the coldest quarter (Bio19)?

It is also quite intriguing to me that five entities are modeled separately based on their taxonomic affiliation (varieties), while genetic analyses only support two clusters (species). Why not be consistent and work with these two clusters only? This would also simplify a lot the message, especially Figures 2, 3 & 4.

Results
They are almost perfect.

Discussion
The parts dedicated to past environments (L384-390, also L407-414) were very interesting to read and well linked with both previous literature and the current results.

But, like in the Introduction, I feel that we miss a general discussion of the implications of this study and its findings for an understanding of the history of the “mexical”. Without such a discussion (to be clear: I am not asking for two sentences, but rather for a full-bodied section), we don’t really see the importance of the study.


Minor comments
L49. What do you mean by ‘the longest mountain’ ? Do you actually mean distance (it would be a bit strange and I would rather refer to area) or do you mean the oldest ?
L58. either ‘the temperate flora’ or ‘temperate floras’
L75. Please write ‘the latter being’
L89. Here the authors should refer to the figure showing the distribution map of the species, possibly figure 1b but it would be even better to have a separate figure showing the geographic distribution of the species, as well as the geographic features evoked in the manuscript (i.e. deserts, mountains ranges, etc.)
L101. Please write ‘that have created the genetic diversity’
L126. You should give references for the 3 genes used here. In particular, it would be interesting to present the two nuclear genes, which will not be familiar to most plant phylogeneticists: are they single copy? How variable are they?
L160. Can the authors justify their choice of a maximum number of K=7 clusters? It seems relatively low compared to the number of named subspecies (which might correspond to geographic clusters).
L204. Why didn’t the authors use records from herbaria in the Southern USA, aren’t they available?
L227. The explanation for this test is interesting: why not explaining this in the introduction?
L366. This is actually a very important point in my opinion, that the authors could bring more in the forefront of their article. Can they report Fst between the two species, average raw genetic distances, time of divergence?

Figure 2: the authors should describe the background map (in shades of grey) that they used: does it show elevation? Also, why not name the geographic bodies mentioned in the text?

---

## Round 0.2 · Minor Revisions

Dear Dr. Moreno-Letelier,

The reviewers found your manuscript was strongly improved and they detected only minor concerns. However, reviewer 3 asks for further details in the analyses. In particular, he/she asks to add multilocus phylogeny at least in the supplementary and he/she doesn't understand why STRUCTURE analysis was stopped at K=7. So, I encourage you to improve the manuscript according to tips of the reviewer and to clarify these two aspects. Please, respond point-to-point to the comments of reviewers to speed up the process of revision.

Once again, thank you for submitting your manuscript to PeerJ and we look forward to receiving your revision.

Sincerely,
Gabriele Casazza

Reviewer 1 ·

Basic reporting

Meets basic reporting standards

Experimental design

No problem

Validity of the findings

Validity of findings is adequate

Additional comments

The manuscript has been significantly improved and I am happy for it to be accepted. Apart from the suggestion to include a very brief description of the conservation status of var. gamboana (if there is space) I only have a few comments (see below):
Line 33 Which genetic cluster? The north or south?
Line 36-37. Best to order these impacts on the species genetic diversity by the order in which they have occurred (if it is not already) and, if there is room, say when they happened (e.g. during Glacials or the Holocene).
Line 65 What is meant by ‘historical relevance’?
Line 174-178 This paragraph is not needed. I suggest deleting it.
Line 247 Was the LIG warmer or wetter than the current interglacial? Given the SDM results for this period some comment should be provided about this.
Line 318 Should ‘Ne’ be italicized?
Line 426 What kind of evidence? Pollen?
Line 443 Change ‘to’ to ‘in the’.
Line 572-573 Like what? Contemporary gene flow?
Figure 1 What is the variety classifications based on? This should be indicated in the Figure 1 description. Also, the north arrow seems too big and out of place. Could it be changed to match the way it is shown in Figure 3?

Reviewer 2 ·

Basic reporting

Thank you for revising the manuscript. I have confirmed all changes related to my previous comments.

Experimental design

Nothing.

Validity of the findings

Nothing.

Additional comments

Nothing.

Reviewer 3 ·

Basic reporting

The manuscript by Martinez de Léon and colleagues presents a clear study of the phylogeography Juniperus deppeana. It follows the standards of peer-reviewed articles (I did not check PeerJ’s specific ones though) and figures are OK.

Experimental design

The work presented here falls within the scope of PeerJ, it is well defined and is both executed and presented rigorously.

Validity of the findings

I trust all analyses and conclusions presented in this work.

Additional comments

I have now assessed the revision of the manuscript by Martinez de Léon and colleagues (I was ‘reviewer 3’ in the previous round of reviews) and I still think that it presents a clear study of the phylogeography of Juniperus deppeana.

Some improvements have been made to the manuscript, which I find now has a broader scope. In particular I appreciate the efforts that have been made to describe the mexical vegetation and to discuss how this study contributes to understand its history.

What still bothers me is the ambiguity between the current taxonomic treatment of these populations and genetic results. The authors are constantly moving between one and the other, sometimes justifying some analyses based on traditional taxonomy, other times on genetic clustering alone (see example below).

All of the points I raised in the first review have been addressed either by making changes to the text or directly in the rebuttal letter. However, I see that none of the analyses have been changed, which I find a bit frustrating. To give specific examples:
(i) why did the authors not include their multilocus phylogeny in the manuscript (or in the supplementary) if they built one? The fact that this tree was poorly resolved is actually a result that should not be hidden I think.
(ii) again, if the aim is to ‘assess[] genetic structure’ (quote from the main text of the manuscript), I don’t understand the justification for stopping the STRUCTURE runs at K=7 because ‘higher values for K when the resulting plot stopped giving additional information and only showed within group structure’ (quote from the rebuttal letter). The authors are precisely answering that these higher numbers of clusters were showing some structure, right? Shall I assume that this structure was deemed irrelevant based on an a priori opinion?

Apart from that I have only two minor points to raise.

MM: please do not write ‘10 non-correlated BIOCLIM variables’. From what the authors responded to my comment on this point in the first submission I understand that some of these 10 variables are most likely highly correlated, but with a correlation coefficient lower than 0.8.

In the supplementary materials, in one instance CHELSA is written ‘CHLESA’

---

## Round 0.3 · Minor Revisions

Dear Dr. Moreno-Letelier,

You addressed the suggestions of the reviewers and the manuscript have been improved. However, before final acceptance, there are still a few points (listed below) to solve.

Thank you for submitting your work to PeerJ.
Sincerely,
Gabriele Casazza

L68. Please briefly explain what the ecological or historical relevance is.
L186. In figure S4.1 are reported values for 10 K. The text or the figure are wrong. Please, check and correct it.
L215. Please, add a comma after “purpose”.
L337. Please round the number to the second decimal place.
L432. Please replace “Our results however,” with “Our results, however,”
L461. Please, “Million” not in capital letter
L511-513. Please consider changing in “Climate conditions during LIG were warmer and wetter than present ones (Berkelhammer et al., 2021), which could help to explain the difference between past and present distribution of J. deppeana.”
Please check for double spaces throughout the text

---

## Round 0.4 · accepted · Accept

Dear Dr. Moreno-Letelier,

I am very pleased to inform you that your paper "Incipient speciation, high genetic diversity, and ecological divergence in the alligator bark juniper suggest complex demographic changes during the Pleistocene" is accepted for publication in the PeerJ. Congratulations!
Thank you for submitting your work to PeerJ.

Sincerely,
Gabriele Casazza